# Modeled buoyancy of eggs and larvae of the deep-sea shrimp *Aristeus antennatus* (Crustacea: Decapoda) in the northwestern Mediterranean Sea

**Morane Clavel-Henry**[1*], **Jordi Solé**[2], **Trond Kristiansen**[3], **Nixon Bahamon**[1,4], **Guiomar Rotllant**[1], **Joan B. Company**[1]

**1** Department of Renewable Marine Resources, Institut de Ciències del Mar—Consejo Superior de Investigaciones Científicas, Barcelona, Spain, **2** Department of Physical and Technological Oceanography, Institut de Ciències del Mar—Consejo Superior de Investigaciones Científicas, Barcelona, Spain, **3** Department of Marine Biogeochemistry and Oceanography, Norwegian Institute for Water Research, Oslo, Norway, **4** Centro de Estudios Avanzados de Blanes—Consejo Superior de Investigaciones Científicas, Blanes, Spain

☯ These authors contributed equally to this work.
* morane@icm.csic.es

**Data Availability Statement:** All relevant data are within the paper and its Supporting Information files.

## Abstract

Information on the buoyancy of eggs and larvae from deep-sea species is rare but necessary for explaining the position of non-swimming larvae in the water column. Due to embryonic morphology and ecology diversities, egg buoyancy has important variations within one species and among other ones. Nevertheless, it has hardly been explored if this buoyancy variability can be a strategy for deep-sea larvae to optimize their transport beyond their spawning areas. In the northwestern Mediterranean Sea, protozoea and mysis larvae of the commercial deep-sea shrimp *Aristeus antennatus* were recently found in upper layers, but to present, earlier stages like eggs and nauplii have not been collected. Using a Lagrangian transport model and larval characteristics, we evaluate the buoyancy and hydrodynamic effects on the transport of *A. antennatus'* larvae in the northwestern Mediterranean Sea. The transport models suggested that 75% of buoyant eggs released between 500 and 800 m depth (i.e., known spawning area), reached the upper water layers (0–75 m depth). Then, according to the modeled larval drifts, three spawning regions were defined in the studied area: 1) the northern part, along a continental margin crossed by large submarine canyons; 2) the central part, with two circular circulation structures (i.e., eddies); and 3) the southern part, with currents flowing through a channel. The number of larvae in the most upper layer (0–5 m depth) was higher if the larval transport model accounted for the ascent of eggs and nauplii (81%) instead of eggs reaching the surface before hatching (50%). The larvae reaching the most water upper layer (0–5 m depth) had higher rates of dispersal than the ones transported below the surface layer (deeper than 5 m depth). The results of larval dispersal simulations have implications for the understanding of *A. antennatus* larval ecology and for management decisions related to the shrimp fisheries in the northwestern Mediterranean Sea.

**Funding:** This work was supported by the Ministerio de Economía, Industria y Competitividad, Gobierno de España (BES-2015-074126) to support her PhD within the CONECTA project (CTM2014-54648-C2-1-R) to MC-H. The funder had no role in study design, data collection and analysis, decision to publish, or preparation of the manuscript.

**Competing interests:** The authors have declared that no competing interests exist.

## Introduction

Numerous species have a pelagic larval cycle which links the spawning places to the recruitment areas. Larval cycle is a relatively short time lapse compared to the life cycle of marine animal, but it is the phase when large dispersions occur [1]. For benthic species, the distribution of the species mostly relies on the transported larvae. Larvae have several mechanisms for positioning themselves in productive and favorable waters that optimize their growth and displacement [2]. According to those mechanisms, the larvae can be retained on the spawning places or connect to other areas that are of high interest for species with high commercial value. For that reason, many studies specifically addressed larval drifts in order to determine the efficiency of the fisheries management [3, 4].

*Aristeus antennatus* is a deep-sea shrimp with a high commercial value in the northwestern Mediterranean Sea. Since 1980, the reproductive cycle, biology, and the temporal and spatial dynamics of *A. antennatus* have been intensively studied based on data from commercial and scientific surveys [5–8]. The acquired knowledge has contributed to shape a local management plan which restricts fishing activity on *A. antennatus* since 2012 for the harbor with the highest landings (Palamós) [9, 10]. The current local management plan was implemented, partly assuming that the protected population of *A. antennatus* would increase the following years on the trawling grounds, and partly based on the idea that recruited juveniles are related to the spawners living on the trawling restricted area.

Nonetheless, like many deep-sea species, knowledge about the early-life stages of *A. antennatus* and their behaviors is scarce. In the case of *A. antennatus*, knowledge can be gathered from various larvae of Dendrobranchiata species, which is the taxonomic family of *A. antennatus*, *e.g.*, larvae molt after the hatching of eggs with the following order: nauplii, protozoea, and mysis. Lecithotrophic stages (eggs and nauplii) of *A. antennatus* have not been observed yet, but its first planktotrophic stage (protozoea) is largely found in the surface layer [11]. Because ripe females spawn on the sea bottom, the embryonic and nauplius stages are assumed to have abilities (buoyancy, swimming ability) that take them to the surface. In [12], an Individual-Based Model (IBM) study showed that the near-bottom circulation has minor effect on passive individual dispersal (encompassing neutral eggs and inactive larvae of *A. antennatus*). The study indicated that near-bottom vertical currents do not advect individuals up to shallower layers, and suggested that the vertical distribution of eggs or larvae can only be explained by their capacity of moving across the water column. To our knowledge, the youngest stage of *A. antennatus*' larvae found in superficial waters is the first substage of protozoea [11, 13]. This finding supports the hypothesis of positive buoyancy in the previous stages of *A. antennatus*' larvae.

Buoyancy adjusts the vertical egg position in the water column by the difference between egg and water densities [14–16]. To date, because either some deep-sea species are gravid (i.e., eggs carried by the spawners) or egg information is unavailable, larval dispersal of deep-sea species rarely accounted for the simulation of the stages that interact with the water density. Within the Dendrobranchiata family or within other deep-sea Decapod species, diverse egg sizes and densities [17] were observed but were not comparable with the assumed characteristics of *A. antennatus*' eggs. In advanced stages of Decapod larvae from various ecosystems (coastal, estuarine, and deep-sea), several patterns of vertical distribution and displacements have also been described [17]. For example, Penaeid nauplii can be found in superficial layers [18, 19] or near the bottom [20]. This biodiversity translates how challenging and delicate is the generalization of the larval behavior to each unknown larval stage of the deep-sea red shrimp.

Although little is known about the *A. antennatus*' larvae, and as for other deep-sea species, it is likely that the vertical position of larvae is determinant for dispersal because the surface

water has stronger velocity and higher temperature than deeper water layers [21, 22]. A number of hydrodynamic processes in the upper waters of the northwestern Mediterranean Sea can disperse or retain the larvae. The main circulation is driven by the Northern Current that flows southwestward between the surface and 250 m depth with maximum velocities around 0.30 m/s [23]. Several eddies circulate in a clockwise or anticlockwise direction and drive the distribution of larvae in the surface layer [24]. Those structures are unstably present during a few weeks and have different sizes and locations [25]. In summer, the Mediterranean Sea is also well-stratified by a thermocline around 15 m deep [26]. This thermocline forms a physical barrier for some small-scale ocean processes such as water mixing [27], and therefore can limit vertical displacement of Decapod larval stages [28, 29].

In this paper, we analyze the impact of buoyancy variability and three-dimensional water mass circulation on the drifts of eggs and larvae between spawning and recruitment areas for the benthic deep-sea shrimp *A. antennatus* species in the northwestern Mediterranean Sea. This work was an opportunity to test for the first time a protocol to define buoyancy values for undescribed larval stages of a deep-sea species and to provide new elements for the management plan makers.

## Material and methods

An Individual-Based Model (IBM) for embryonic and larval stages of the deep-sea red shrimp (*A. antennatus*) was implemented with a Lagrangian particle-tracking framework. It simulated the early life-cycle behavior and dispersal patterns of shrimp larvae using 3D hydrodynamic model outputs of the northwestern (NW) Mediterranean Sea.

### The hydrodynamic model

A climatological simulation of the NW Mediterranean Sea hydrodynamics was run using the Regional Ocean Modeling System (ROMS; [30]), a free-surface, terrain-following, primitive equation ocean model. A Mellor-Yamada 2.5 turbulence closure scheme is used for subgrid-scale mixing in the simulation [31]. Different models (IBM in [12], and a spatiotemporal food web model (Ecopath with Ecosim) in [32]) have successfully used the daily outputs of the ROMS model in the NW Mediterranean Sea and on the Valencian Gulf. The simulation domain ranged from 38° N to 43.69° N and from 0.65° W to 6.08° E (Fig 1), with a grid spacing of 2 km (with 256 x 384 grid points horizontally). The vertical domain is discretized using 40 vertical levels with a finer resolution near the surface (surface layer thickness between 0.49 m and 5.91 m). ROMS is built forced by a high resolution and accurate bathymetry of the western Mediterranean basin, which is fundamental for the drift study of *A. antennatus'* eggs and larvae because adults are benthic and females spawn around 800 m depth [33].

The ROMS outputs are validated in [12] and provided realistic products of the hydrodynamic and hydrographic characteristics (current velocities, salinity, and temperature) of the NW Mediterranean Sea. The upper water layers were characterized by the main southward current (i.e., the Northern Current) following the eastern coast of Spain and by several mesoscale circulations, like eddies. We estimated eddy size and center over monthly-averaged ROMS current velocities field with an algorithm described in [34]. Seawater density is a derived product of the salinity and temperature (equation 13 in [35]). On the horizontal dimension, average estimated temperature, salinity, and seawater density in summer (i.e., July to September) are 27.6°C, 37.5 PSU, and 1026.3 kg/m$^3$ at the surface and 13.1°C, 38.2 PSU and 1035.35 kg/m$^3$, at bottom, respectively. Those values are in the range of seawater measurements carried out in the NW Mediterranean Sea [23, 36].

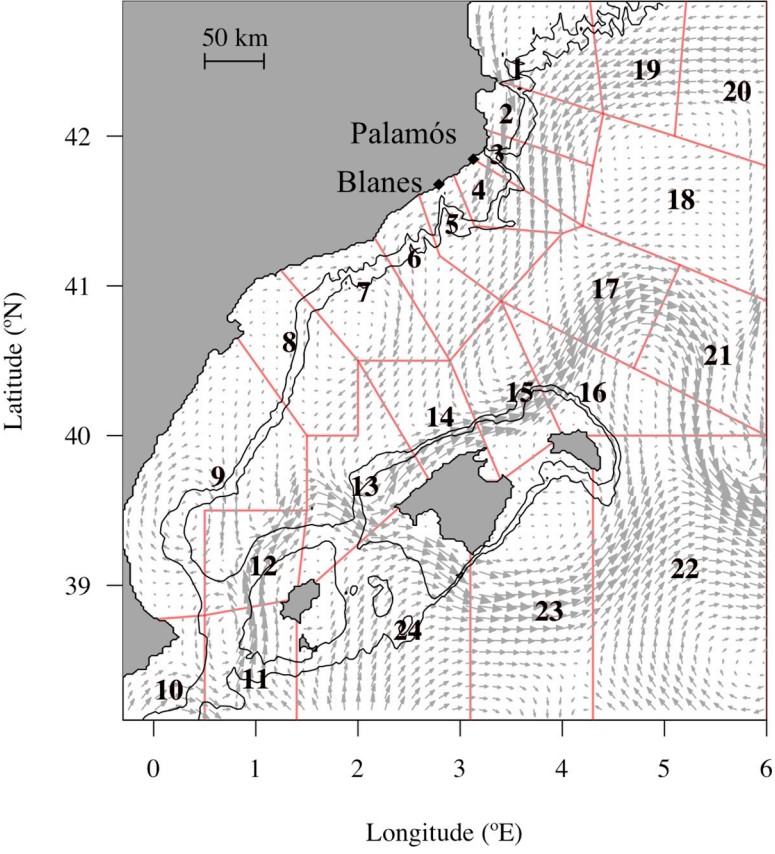

**Fig 1. Release and settlement zones defined in the northwestern Mediterranean Sea for this study.** The domain is divided by release zones (polygons from 1 to 12) and settlement zones (polygons from 1 to 24) along the continental margins of the management unit called Geographical Sub Area 6 (GSA 6) from the Food and Agriculture Organization. Eggs were spawned in the release zones at the seafloor between the 500 m and 800 m isobaths (black lines) where the mature female shrimp aggregates. All the zones (polygons from 1 to 24) are considered potential settlement zones of the deep-sea red shrimp. Arrows represent the average surface current (m/s) over July from the hydrodynamic model used in the study.

## The Individual-Based Model

The Lagrangian drift of individuals was calculated using the Python framework OpenDrift ([37]; available on https://github.com/OpenDrift). The equation for the advection of individuals in the three dimensions is resolved by a fourth-order Runge-Kutta scheme. Particles were defined as 'stranded' when they came into contact with the coastline. The modules for turbulence and buoyancy were activated by the modeler. The number of particles, the release coordinates and the duration of the drift setting the initial conditions of the Individual-Based Model are also described in the following sections.

**Horizontal and vertical components.** The trajectories of individuals, representing virtual embryonic or larval stages, were based on the following equation:

$$\frac{dX}{dt} = U(X, t) + B(X, t) + D_h(X, t) + D_v(X, t) \tag{1}$$

where $dX/dt$ is the 3D displacement of the individuals from their geographical and vertical position $X$ on a time step $dt$; $U$ is the advection component composed by the meridional, zonal

and vertical velocities from ROMS; *B* is the biological component, and $D_h$ and $D_v$ are the small-scaled turbulent velocities in the horizontal and vertical dimensions, respectively.

The vertical velocity of individuals due to turbulent diffusivity ($D_v$) is adjusted by the random displacement scheme [38] with an internal time step of 20 s and a 1 m vertical resolution. Values of $K_v$ were provided from the coefficient of salinity vertical diffusion from ROMS outputs. In our simulation, the vertical mixing was activated until individuals crossed the oceanic surface boundary layer (around 15 m deep [26]). Because the Mediterranean Sea is well-stratified in summer, above the oceanic surface boundary layer, the turbulence was accounted by simulating the horizontal mixing.

The horizontal velocity due to turbulent diffusivity was computed by a random walk scheme:

$$D_h = rand \cdot \sqrt{2 \cdot K_h} \cdot dt \qquad (2)$$

with *rand* representing a random value sampled in a Gaussian distribution G(0,1), *dt* the time step, and $K_h$ the horizontal diffusivity coefficient. For $K_h$, we used the yearly constant and average value of 10 m$^2$/s estimated by [39] in the Western Mediterranean Sea from a hydrodynamic model with similar horizontal resolution than ours.

The biological component *B(X, t)* represented the vertical terminal velocity due to the buoyancy force adjusted to the seawater density by the equilibrium of the Archimedes, gravitational, and friction forces under laminar flow (Reynolds number < 0.5) on a spherical object characterized by diameter and density. For the present study, this object size was randomly sampled in a Gaussian distribution with the average *A. antennatus'* egg diameter of 3.3*10$^{-4}$ m and its standard deviation at 0.45 *10$^{-4}$ m [6]. The density of the sphere was selected after the analysis described in the sections below. The terminal velocity is computed by the equation provided in [40], or by Dallavalle equations [41] if the flow is transient (Reynolds number > 0.5).

**Adaptation of the IBM to the shrimp larval cycle.** Following the temporal and spatial spawners distribution, virtual eggs [6, 12, 33] were set to be released in summer at midnight [42] on the bottom of the NW Mediterranean continental slope between 500 m and 800 m isobaths.

The IBM included activation and deactivation of the buoyancy force according to the individual stage during the drifts. The buoyancy module was initially activated at the release of the eggs. The eggs were positively buoyant by assuming that on the one hand, young larvae of *A. antennatus* are found in the surface layer [11, 13] and on the other hand, if the buoyancy is neutral or negative, the swimming abilities of the nauplius could not explain an average vertical rise on 600 m. When individuals reached a non-buoyant stage, the module was deactivated. For simplification, and due to lack of knowledge, we neglected the temporal change of egg size and density.

The drift duration was the sum of the pelagic egg and larval stage durations approximated by the temperature-dependent Pelagic Propagule Duration [43, 44]. Due to missing information regarding the embryonic and larval stages of *A. antennatus*, we assumed that its early-life development duration was similar to the near taxonomic Penaeid species. We reviewed 72 research articles (see S2 File) in which the larval stage duration of 42 Penaeid species was associated with the rearing water temperature. Then, we fitted a multiple linear model on those data to estimate *A. antennatus* larval stage durations (*D*, in day) according to the seawater temperature, such as:

$$\log(D) = T + Stage + \varepsilon, \qquad (3)$$

where $T$ is the rearing water temperature; *Stage*, a categorical variables characterizing the Penaeid larval stage and $\varepsilon$ a random error. This model, whose initial assumptions were verified with a coefficient of determination $R^2 = 96\%$, had the following shape:

$$D_{stage} = \exp(-0.072 * T + \begin{array}{ll} 1.51, & if\ stage = eggs \\ 2.66, & if\ stage = nauplius \\ 3.68, & if\ stage = protozoea \\ 3.64, & if\ stage = mysis \end{array}). \tag{4}$$

We estimated egg stage duration before the beginning of the drift simulations and larval stage duration was estimated each time an individual molted into the next stage (e.g., from nauplius to protozoea). The water temperature involved in Eq 4 was extracted at embryonic or larval position from ROMS model. Molting was allowed only if the simulation time was bigger than the cumulated duration of the previous and current stages.

**Number of particles.** The methodology of [45] adjusted in [12] was used to determine the lowest number of individuals to be released and to guarantee that 95% of the dispersal variability is considered in our results. It is based on the average of the Fraction of Unexplained Variance [45], which is got by extracting randomly 100 subsamples of N drifts from 350,000 simulated trajectories and by computing the cross-correlation between the 100 subsamples. The subsample size N ranged between 1,000 and 300,000. The drifts of 350,000 individuals directly started at the surface with underneath sea bottom estimated between 500 and 800 m. Assuming to catch the full range of dispersal possibilities for larvae, the 350,000 drifts lasted the maximum PPD (38.8 days). The PPD was predicted from the relationship as described in the previous section, considering the coldest water near the bottom of the studied area (12.6°C) where eggs are spawned. Turbulent diffusivity was as well included for the drifts of 350,000 individuals. To avoid the underestimation of needed individual number, the 350,000 drifts also included a horizontal turbulent random walk with the maximum value of the coefficient $K_h$ estimated at 100 m²/s in [39] instead of its average at 10 m²/s. Finally, the Fraction of Unexplained Variance was lower than 5% if 50,000 individuals are used (S1 Fig) to simulate the larval dispersal.

## Larval dispersions

Modeled dispersion started with buoyant stages having predicted densities (in kg/m³) to rise toward upper water layers. When the stages were no more buoyant, the individuals were neutrally advected by the currents of the reached water mass layer.

**Preliminary IBMs.** Beforehand, considering the few biological data available on taxonomically close species to *A. antennatus* and the important differences in larval ecology of deep-sea species (see S1 File), the density for the eggs reaching the surface needed to be estimated. This estimation was carried out relying on physical explanation (e.g., forces applied on the particles) to model the egg rise instead of unknown biological values.

To obtain the estimation (hence called optimal buoyancy), the depth reached by the individuals was analyzed when the buoyant stages ended. In the early summer (July 1), 50,000 particles were released and tracked up to the end of the nauplius stage (~ 7.5 days). The Lagrangian drift simulations of eggs were repeated for every 10 kg/m³ increment, between 800 and 1030 kg/m³. Then, for each particle reaching the surface (0–5 m depth layer), their densest value was kept. It represented the threshold where a denser value would not make the individual reach the surface and a lighter value would likely lead the individual to the surface. The average and standard deviation of the highest density values were kept to implement the drift simulation of the particles.

Other averages and standard deviations of the egg density were estimated by modifying one parameter in the configuration of the previous preliminary IBM. Independently, we computed those parameters i) for individuals reaching the upper boundary of the seasonal thermocline (about 15 m under the surface; [26]) instead of the surface layer, ii) when the buoyancy was applied during egg and nauplius stages, instead of egg stage only, iii) when the turbulence was activated, and iv) when the drift began at late summer (September 1). In case i), the method used to find the seasonal thermocline depth, we used the maximum slope of difference in the temperature profile during summer days [46]. In case ii), the particles were tracked up to the end of the nauplius stage (~7.5 days). In total, the preliminary experiment consisted of the analysis of simulations made with 24 values of egg density for six configurations of IBM (S1 Table).

**Runs and sensitivity scenarios of IBMs.** We used six scenarios with different configurations of IBM. Sensitivity tests were performed introducing small or high variability due to the buoyant larvae or the hydrodynamics changes (Table 1). The scenario of reference $IBM_0$ started in early summer at the beginning of the peak period of spawning and with buoyant eggs. We modified the average egg density to lead the individuals to different depths (surface or mixed layer depth), to use different buoyant stages (eggs or eggs + nauplii), to include the vertical and horizontal turbulent components, and to begin the drift simulation in either early or late summer ($1^{st}$ July or $1^{st}$ September). In the last scenario $IBM_{Hot}$, the temperature in the upper 200 meters was incremented by 0.4˚C, corresponding to the thermal increase expected in the upper layer of the western Mediterranean Sea over a decade [47].

Those scenarios were operated to consider the factors associated with larval ecology ($IBM_0$, $IBM_{PZ}$, and $IBM_{MLD}$) and to the hydrodynamics ($IBM_{LS}$, $IBM_{Turb}$, $IBM_{Hot}$). Through $IBM_{MLD}$, we explored the fact that the thermocline created a physical barrier for the larvae [28, 29]. In $IBM_{PZ,}$ we explored the possibility that buoyancy was still taken into account in the vertical movements of nauplius after hatching. The swimming abilities of the spheroid nauplius of Penaeid can hardly lead them to the surface [48, 49]. Yet, the nauplius stages of some coastal shrimps were found in the superficial layer [50], which means that an underlying process helps them to rise in the water column. Finally, the last scenarios allowed making a sensitivity analysis on the turbulent mesoscale impact on the main advection with the turbulent component ($IBM_{Turb}$), and on the temporal changes in the physical environment (late summer hydrodynamic circulation) on the drifts ($IBM_{LS}$). $IBM_{hot}$ was used to estimate the impact of a scenario of climate change on the larval drifts, through the rising of water temperature above average, which is expected to have a major impact on the larval duration and in seawater density.

**Table 1. Configurations of IBM scenarios for egg and larval drifts.**

| IBM Scenario | Buoyant larval stages | Displacement schemes | Depth reached by buoyant stages | Release event | Density (kg/m$^3$) |
|---|---|---|---|---|---|
| $IBM_0$ | Eggs | U(X, t) + B(X, t) | Surface | July 1 | 884 ± 36 |
| $IBM_{PZ}$ | *Eggs + Nauplii* | U(X, t) + B(X, t) | Surface | July 1 | 979± 14 |
| $IBM_{Turb}$ | Eggs | U(X, t) + B(X, t) + *Dh(X, t) + Dv(X, t)* | Surface | July 1 | 885 ± 36 |
| $IBM_{MLD}$ | Eggs | U(X, t) + B(X, t) | *15 m* | July 1 | 887 ± 36 |
| $IBM_{LS}$ | Eggs | U(X, t) + B(X, t) | Surface | *September 1* | 882 ± 36 |
| $IBM_{Hot}$ | Eggs | U(X, t) + B(X, t) | *Surface** | July 1 | 884 ± 36 |

*U*, advective component by the meridional, zonal and vertical velocities; *B*, velocity due to the buoyancy force; $D_v$ and $D_h$, the velocity due to the vertical and horizontal diffusivity; *PZ*, Protozoea; *Turb*, Turbulence; *MLD*, Mixed Layer Depth; *LS*, Late Summer. Changes in parameterization with respect to the base scenario $IBM_0$ are in italics.

* Temperature in the 0–200 m layer has been increased by 0.4˚C.

For each scenario, the density (kg/m$^3$) of the individuals was randomly sampled from a Gaussian distribution (Fig 2), using the average and standard deviation density values from the preliminary analysis. The simulation duration lasted the time that an egg developed into early juvenile according to the water temperature when larvae molted. The time step to advect individuals was one hour. The characteristics of the individuals (density, egg size, stages, duration of the stage) and their spatial position (latitude, longitude, and depth) were saved on a daily basis for further analyses.

### Analysis of the drift simulations

The water characteristics during the drifts, the drift duration, the vertical rise of the individuals from spawning depths and the traveled distance were used to analyze the simulated drifts from the six scenarios. Distance $d$ between two geographical positions given by the coordinates (X, Y) was computed by the haversine formula of great-circle distances;

$$d = arcos[\sin(Y_1) \cdot \sin(Y_2) + \cos(Y_1) \cdot \cos(Y_2) \cdot \cos(X_1 - X_2)] \cdot R, \qquad (5)$$

with $R$ as the radius of Earth. The drift distances corresponded to the aggregated distance during the simulation. The straight distance corresponded to the distance between the beginning and the end of the drifts.

In our study, we also analyzed the individual dispersals within and among areas with the connectivity matrices. Therefore, the NW Mediterranean Sea domain was divided into 24 zones (Fig 1) shaped by the main structures of topography like the Eivissa channel, the submarine canyons (Cap de Creus, Palamós, Blanes), and the gulfs (Valencian Gulf or Lion Gulf). In creating the zones we took account of the zoning criteria provided by the General Fisheries Commission for the Mediterranean (Geographical Sub-Areas 5, 6 and 7). Then, the connectivity rates between one release zone $i$ and one settlement zone $j$ were computed from the ratio of individuals in the settlement zone to the initial individuals from their release zone $N_{j|i}/N_i$. The release zones were the zones 1 to 12 while the settlement zones included all the 24 zones

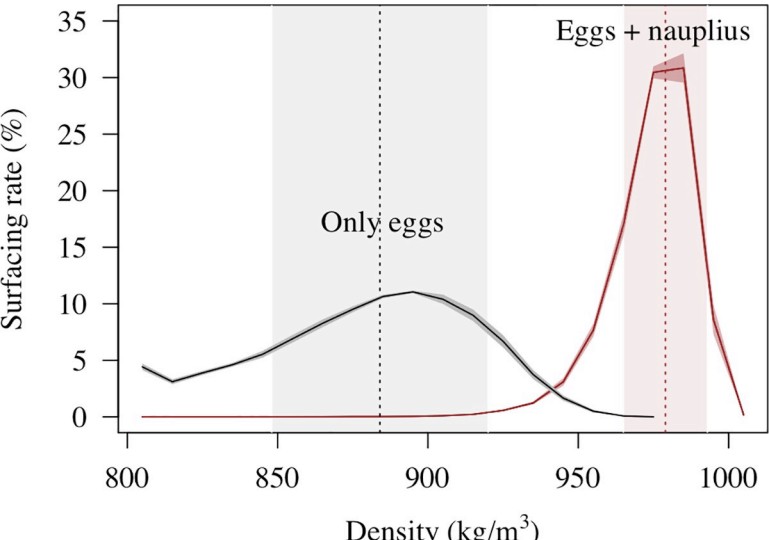

**Fig 2. Modeled surfacing rates for eggs and nauplii of *A. antennatus*.** Surfacing rate of buoyant eggs (black line) or buoyant eggs and nauplii (brown line) related to different tested densities (kg/m$^3$). The average (vertical dashed lines) and standard deviation (colored ranges along the X-axis) of the water densities were used in different scenarios of Individual-Based Model (IBM). See details in Table 1.

(Fig 1). In order to focus on the most relevant results, statistical tests and principal component analyses were implemented with averaged and scaled data with the basic packages of R.

## Results

The simulated larval drifts significantly varied with the spawning places, the number of buoyant stages, the spawning period and the depth layer reached by buoyant phase.

### Larval drifts in three spawning regions

The temporal hydrodynamic and individual buoyancy variations allowed setting relevant scenarios to use for connectivity analysis. Using the characteristics of the larval drifts ant the environmental influences (PPD, drift distances, water current, and water temperature) by scenarios, a Principal Component Analysis (PCA) differentiated three clusters of scenarios (Fig 3). One gathered all the IBM parameterized by small changes with respect to $IBM_0$ (i.e., $IBM_{Turb}$, $IBM_{hot}$, and $IBM_{MLD}$). The non-parametric Kruskal-Wallis test revealed that their PPD and drift distances were significantly different (p-values > 0.05). Nevertheless, their larval drifts expanded over a longer period (25.6 days) and over longer distances (118 km) even though larvae drifted at the thermocline depth ($IBM_{MLD}$), or within small-turbulent water ($IBM_{Turb}$), or in warmer water ($IBM_{hot}$). The two other clusters were defined by the scenario $IBM_{LS}$, in which larval drifts traveled the smallest distances (93 km), and by the scenario

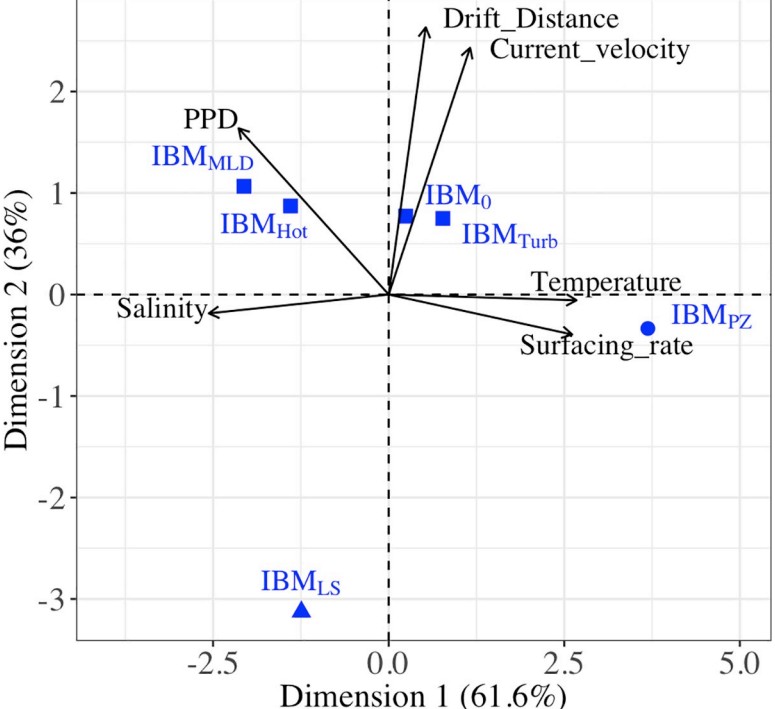

**Fig 3. Scenarios of Individuals-Based Models related to the larval drift characteristics and the environmental influences.** The different scenarios of Individuals-Based Models used in this study are represented by blue symbols (circle, triangle or square) and visualized with the characteristics of larval drift and environmental influences (arrow). In the Principal Component Analysis (PCA), the correlation among the characteristics is indicated by the angle between arrows (i.e., an angle of 90˚ indicates no correlation and an angle of 180˚ indicates a negative correlation). Scenarios were correlated to the larval drift characteristics and environmental influences by the closeness of their Cartesian coordinates. The PCA grouped $IBM_0$, $IBM_{MLD}$, $IBM_{Hot}$, and $IBM_{Turb}$ (square), and separated $IBM_{PZ}$ (circle) and $IBM_{LS}$ (triangle). IBM scenarios are described in Table 1.

$IBM_{PZ}$, in which the surfacing of larvae had higher rates (80%) and PPDs were the shortest (23.2 days). The present analysis focused on the drifts simulated in the scenarios $IBM_0$, $IBM_{PZ}$ and $IBM_{LS}$ that were separated into distinct clusters and had important differences in their drift characteristics (S3 Fig).

Generally, the larval drifts had similar trends for individuals released in specific latitudinal ranges of the NW Mediterranean Sea. For each scenario previously selected, a PCA (Fig 4) was carried out on the larval drift characteristics and environmental influences by release zones, gathering them into three regions. Besides, those three regions were driven by the same variables in each scenario. The region 1 gathered larvae released from zones 1 to 4 (approximately 41.39°N– 42.74°N) with the longest PPD and the straightest transports. The region 2 gathered individuals released between zones 5 and 10 (approximately 38.86°N– 41.39°N), which were abundant in the surface water but drifted the shortest distances. The region 3 gathered the individuals from the zones 11 and 12, and overlapped each side of the Eivissa Channel. Those individuals drifted the furthest and had low surfacing rates.

In each three regions, the individual drifts followed different circulation patterns. In region 1, Fig 5 shows an expanded and linear distribution of the individuals towards the Southwest. The dispersal of individuals from the region 2 shaped different eddies, which are almost-circular structures of the hydrodynamics. In all scenarios, those eddy-like structures were localized where the Gulf of Valencia widens (0.5° E; 39.5° N, and 2° E; 40.5° N). These eddies had only differences in their average radius and the position of their center changed between the early and late summer releases ($IBM_0$ and $IBM_{LS}$). For instance, these eddies get 6 to 8 km larger in $IBM_{LS}$ (S2 Table) and their center displaced 9.5 km to 14.7 km further (S2 Table) than eddies in $IBM_0$. Last, individuals from region 3 were affected by the presence of an eddy with a center averagely positioning in the Eivissa channel (0.72° E; 38.70°N, S2 Table) in early summer ($IBM_0$).

## Advantage of two buoyant stages in the larval drifts

The highest surfacing rate regardless of the regions was when the ascent of larvae occurred with two buoyant stages. With an average density of 979 kg/m$^3$ in $IBM_{PZ}$, 93 kg/m$^3$ denser than in $IBM_0$, 81% of the eggs and nauplii rose to the shallower water layers, while in $IBM_0$, 52% of the eggs reached the surface layer (Fig 6). Overall, the average time for the ascent of two buoyant stages ($IBM_{PZ}$) lasted 7.5 days (i.e., 5.5 days longer than in $IBM_0$). Nevertheless, at the end of the simulations in $IBM_{PZ}$ and for individuals at the surface, the drift durations increased by approximately two days and the drift distances were hardly 3.6 and 4.5 km longer (regions 1 and 2) or 12.6 km shorter (region 3) than in $IBM_0$ (Table 2).

The time lag between early and late arrival of individuals at the surface was small enough for not implying important divergences in the drift characteristics. Indeed, in the two scenarios $IBM_0$ and $IBM_{PZ}$, the surfaced larvae were not spatially exposed to different currents. When nauplii reached the upper 5 m layers in $IBM_{PZ}$, they were 20.5 km away from the location where the eggs have surfaced in $IBM_0$. However, this distance was mostly kept to the nearest kilometer between the individuals of $IBM_0$ with the same age (7.5 days) than the surfaced individuals of $IBM_{PZ}$. It illustrated that while the buoyant nauplii in $IBM_{PZ}$ were still rising in the water column, the surface current in which the individuals from $IBM_0$ were advected, had not important amplitude or direction changes.

The overall consequences from tardive and abundant individuals at the surface were that the connectivity and retention strengthened between and within zones of the NW Mediterranean Sea. In both $IBM_0$ and $IBM_{PZ}$, the same settlement zones were connected but with different amplitudes (Fig 6). First, the dispersal rate between the release zones from the region 1 (zones 1 to 5) and from Blanes canyon and its neighboring zones (zones 5 to 7) rose from

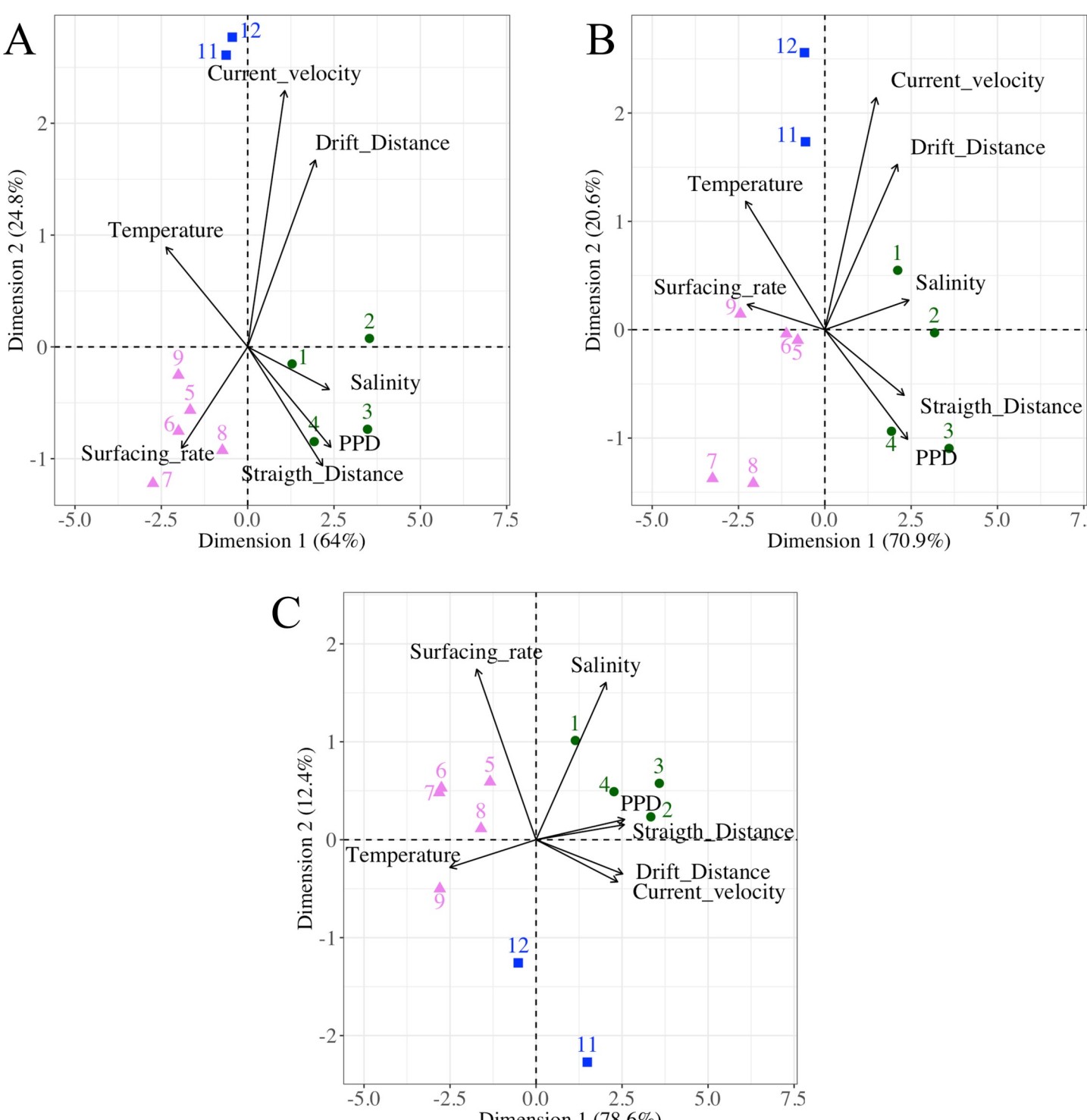

**Fig 4. Larval drift characteristics and the environmental influences of the three most distinct IBM scenarios.** Release zones (colored symbols and numbers) linked to characteristics of the larval drift (arrows) in a Principal Component Analysis for A) the base scenario $IBM_0$, B) the scenario with the buoyant stages up to Protozoea $IBM_{PZ}$, and C) the scenario initialized at late summer $IBM_{LS}$. The correlation among the characteristics of the larval drift is indicated by the angle between arrows (i.e., an angle of 90° indicates no correlation and an angle of 180° indicates a negative correlation). Release zones were correlated to the larval drift characteristics and environmental influences by the closeness of their Cartesian coordinates. The PCA grouped three regions: the release zones 1–4 (full green circle), the release zones 5–9 (full pink triangle) and the release zones 11 and 12 (full blue square). IBM scenarios are described in Table 1.

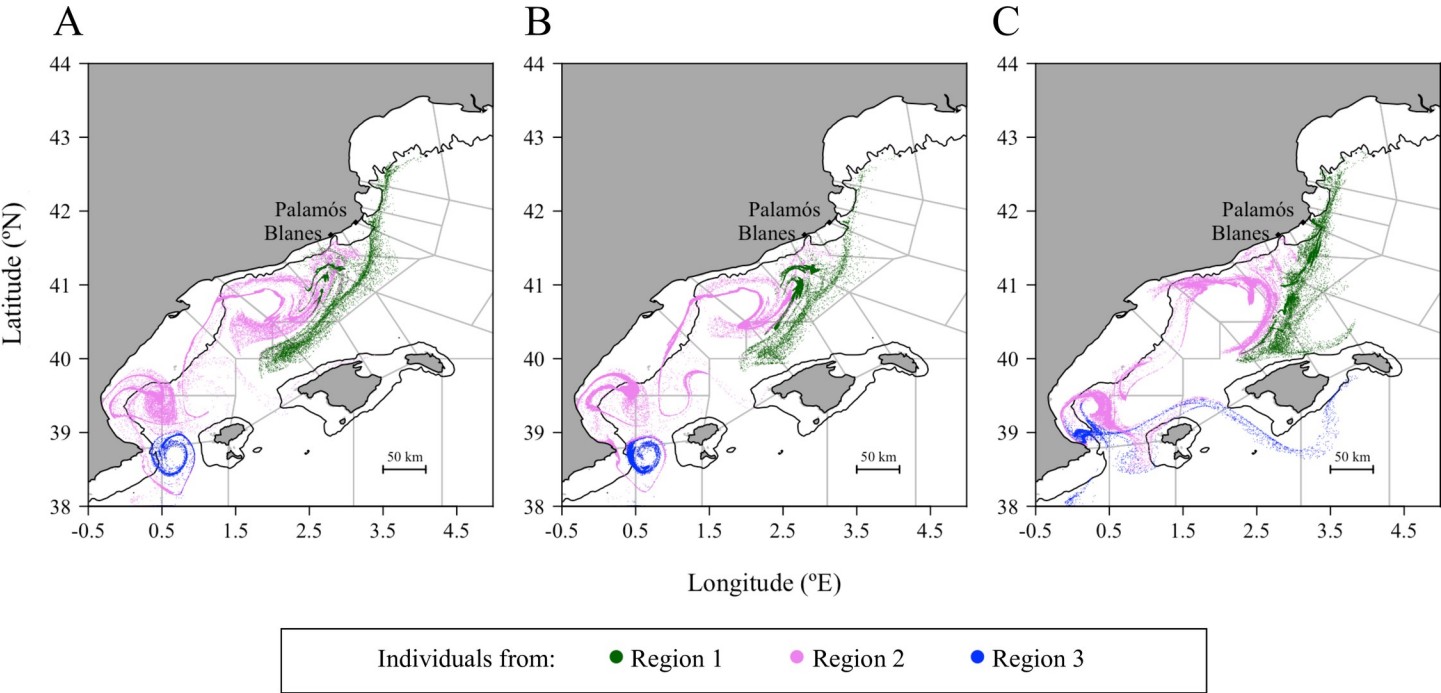

**Fig 5. Settlement position of early juveniles at the end of the simulated drifts by regions and by the three selected IBM scenarios.** The early juveniles are positioned after the simulation with A) the base scenario IBM$_0$, B) the scenario with the buoyant stages up to Protozoea IBM$_{PZ}$, and C) the scenario initialized at late summer IBM$_{LS}$. The regions are defined according to the results of a Principal Component Analysis (Fig 4). The black line represents the shelf break at 200 m. IBM scenarios are described in Table 1.

42.6% in IBM$_0$ to 56.7% in IBM$_{PZ}$. Second, in the Gulf of Valencia's zones (zones 7 to 9), corresponding to the south of region 2, the retention rates were from 33.4% to 43.6% higher in IBM$_{PZ}$ than IBM$_0$. Last, each side of the Eivissa Channel (zone 11 and 12 in the region 3) connected with the other side if the surfacing was earlier like in IBM$_0$. Indeed, in this scenario, the

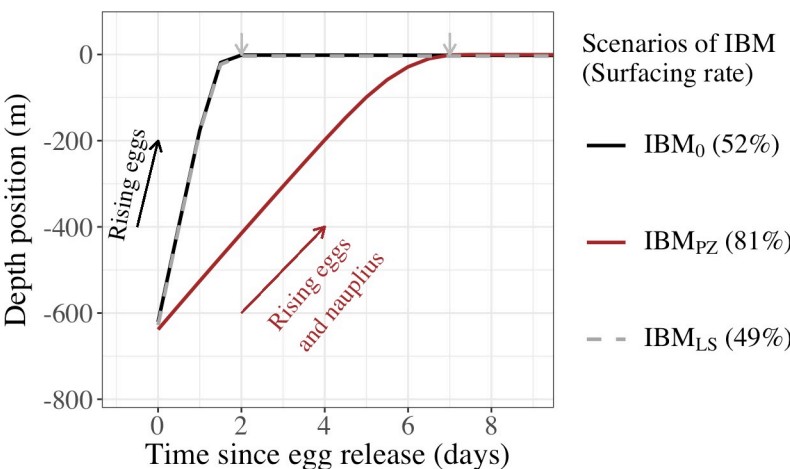

**Fig 6. Depth position of eggs (IBM$_0$ and IBM$_{LS}$), and eggs and nauplius (IBM$_{PZ}$) after spawning.** The depth position is recorded from the release day (Time = 0) to the end of the buoyant stages (grey arrow) in three Individual-Based Models. In IBM$_0$ (black line) and IBM$_{LS}$ (grey dashed line), eggs were spawn at early and late summer, respectively, and buoyant eggs rose during 2 days. In IBM$_{PZ}$ (brown line), eggs and nauplii were buoyant and rose in the water column during 7.5 days.

**Table 2. Characteristics of the larval drifts by the three selected IBM scenarios, by regions of the NW Mediterranean Sea and by depth of drifts.**

| IBM Scenario | Region | Release zones | Surfacing rate | Drift in the 0–5 m depth water layer | | | Drift in the water layer below 0–5 m depth | | |
|---|---|---|---|---|---|---|---|---|---|
| | | | | T | PPD | Drift | T | PPD | Drift |
| IBM$_0$ | 1 | 1–4 | 48.0 | 24.5 | 20.3 | 153.5 | 15.7 | 32.6 | 162 |
| | 2 | 5–9 | 55.6 | 25.9 | 17.6 | 80.1 | 16.5 | 31.2 | 105 |
| | 3 | 11–12 | 43.8 | 27 | 16.3 | 174.5 | 17.9 | 28.9 | 170.6 |
| IBM$_{PZ}$ | 1 | 1–4 | 70.9 | 23.7 | 22.4 | 158 | 16.3 | 31.9 | 173.1 |
| | 2 | 5–9 | 85.8 | 24.9 | 20.1 | 83.7 | 17.2 | 30.3 | 102 |
| | 3 | 11–12 | 77.4 | 25.8 | 19.0 | 161.9 | 17.9 | 29.5 | 168.9 |
| IBM$_{LS}$ | 1 | 1–4 | 45.4 | 22.4 | 20.1 | 147.9 | 16.5 | 30.8 | 145.1 |
| | 2 | 5–9 | 55.3 | 24.2 | 17.2 | 54 | 17.7 | 28.6 | 79.8 |
| | 3 | 11–12 | 26.0 | 25.2 | 15.7 | 57.7 | 18.4 | 27.6 | 113 |

PPD, Pelagic Propagule Duration (in days); T, the average seawater temperature (in ˚C); Drift, the drift distance (in km); Surfacing rate, the percentage of individuals reaching the surface (0–5 m layer); PZ, Protozoea; LS; Late Summer. The regions 1–3 are defined in Fig 4.

exchange of surfaced individuals between the two sides was bidirectional, with a tendency for individuals to cross northwardly the channel. A rate of 21.3% individuals from zone 11 connected to the northern zones of the Eivissa Channel (zone 9 and zone 12) and 11.5% individuals from the zone 12 crossed the channel and arrived on the southern zones of the Eivissa Channel (zones 10 and 11). While in IBM$_{PZ}$, the exchange of late surfaced individuals across the channel was unidirectional and it was done by the northern zones of the channel. Under the influence of an eddy (Fig 4), the southern zones (zone 11 and 10) of the Eivissa Channel received 63.2% individuals from zone 12 and zone 11 kept 61.4% individuals (or 55.6% individuals more than in IBM$_0$).

## Larval drifts after early and late summer spawning

The present results distinguished the circulation in the water from the northernmost region (region 1) from the southern and warmer regions (regions 2 and 3). The drifts at the surface were exposed to higher temporal variability in the circulation fields if they started in the warmer regions of the NW Mediterranean Sea. In IBM$_{LS}$, 49% of individuals (i.e., 3% fewer than in scenario IBM$_0$) rose to the surface, and drifted 24.1 days (i.e., around one day less than in scenario IBM$_0$). Yet, as shown in Fig 3, individuals from IBM$_{LS}$ traveled less. The temporal current changes were measured with the differences in drifted distance per day in scenario IBM$_{LS}$. In region 2, surfaced larvae traveled 26 km less than in IBM$_0$ (i.e., 3.1 km/day instead of 4.6 km/day) and in region 3, surfaced individuals traveled 96.4 km less (i.e., 3.7 km/day instead of 10.7 km/day). In contrast, the individuals in the colder region 1 had similar drifting velocity in IBM$_{LS}$ (i.e., 7.4 km/day) and in IBM$_0$ (i.e., 7.6 km/day).

Besides the temporal changes in the intensity of the circulation fields, two circular structures in the NW Mediterranean Sea boosted the dispersal rates of the surface individuals. First, the diameter of an eddy-like structure between the latitudes 41˚N and 42˚N had modified the surface connectivity of individuals toward the Balearic Islands. In IBM$_{LS}$, the north side of the Majorca Island (biggest island; zone 13 and 14) received 40.9% and 37.9% individuals from the Blanes canyon (zone 4) and its southern zone (zone 5 in Fig 7), while in IBM$_0$, the reception rate of the individuals was lower than 3.5%. This new connection leaned on the changes in the current direction at the end of the summer and the size of the eddy-like structure. Individuals

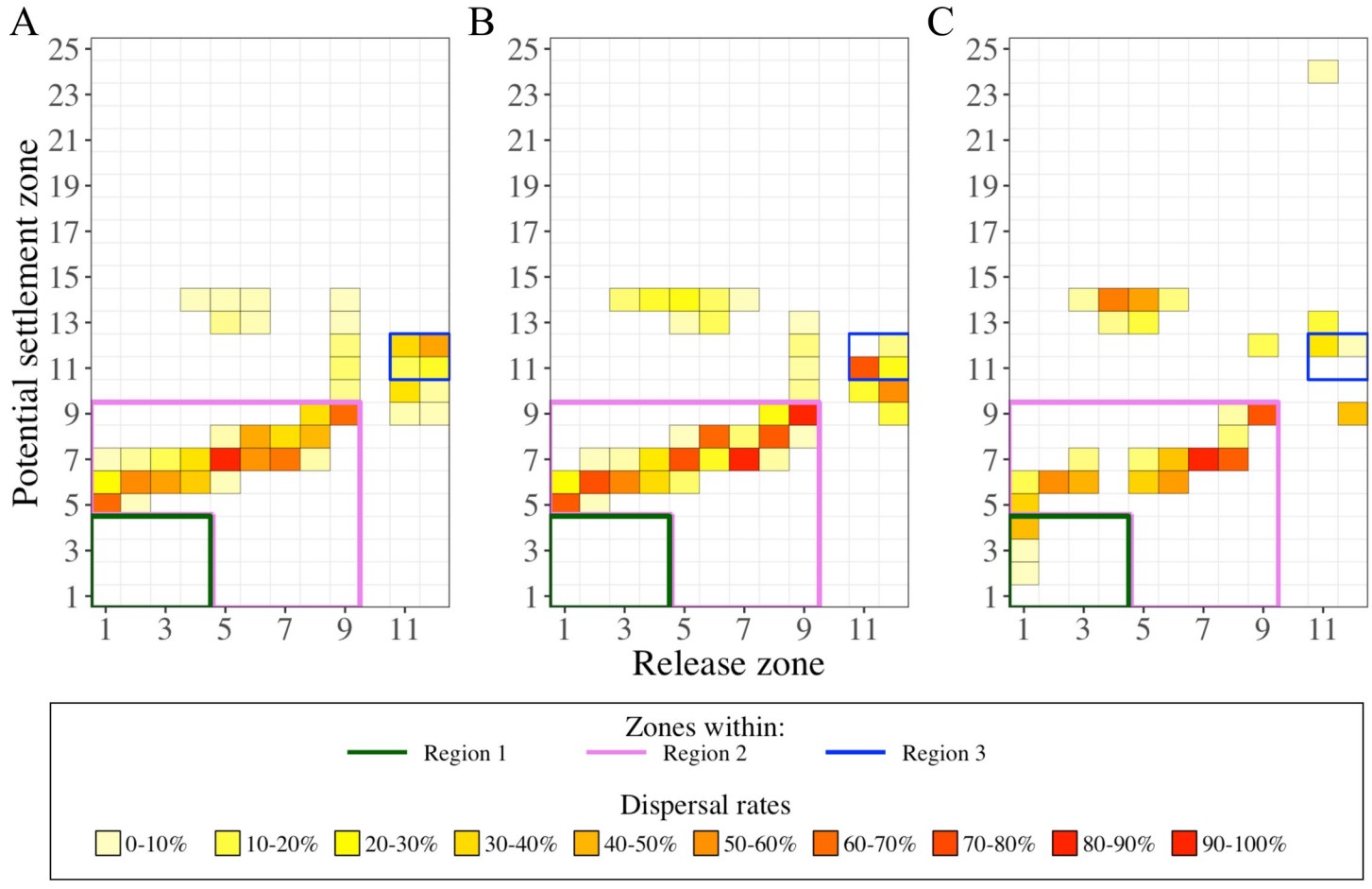

**Fig 7. Dispersal rate in the upper water layers (0–5 m) by the three selected IBM scenarios.** Dispersal rates were calculated from release zones (X-axis) to potential settlement zones (Y-axis) from simulated larval drifts in A) the base scenario $IBM_0$, B) the scenario with the buoyant stages up to Protozoea $IBM_{PZ}$, and C) the scenario initialized at late summer $IBM_{LS}$. For clarity purposes, we contoured the dispersal rates in regions 1 (green lines), 2 (pink lines) and 3 (blue lines). Identification of zones and regions as defined in Figs 1 and 4, respectively. Code color is related to the dispersal rate (%). The sum of the dispersal rates over a column gives the percentage of individuals in the surface layer.

from the region 1 traveled a similar distance per day in $IBM_0$ and $IBM_{LS}$ (7.5 km/day and 7.1 km/day respectively), but the circulation pattern had an angle closer to the south direction, which advected the individuals more southwardly in $IBM_{LS}$ (26° clockwise from geographical South) than in $IBM_0$ (36° clockwise from geographical South). Fig 5 suggests that the eddy-like circulation over the north of the region 2 qualitatively had a bigger diameter. Second, the transport of surfaced individuals across the Eivissa Channel was limited without eddy structure. Indeed, the absence of an eddy in the Eivissa Channel (Fig 5C) paired with the fact that all surfaced individuals from zone 11 were transported unidirectionally across the Eivissa Channel. Additionally, most of the surfaced individuals from zone 12 mostly (25.2%) were advected in the eddy-like structure located at the south of the region 2.

## Larval drifts in different depths of water layer

A part of the individuals did not reach the surface because of the variability in egg characteristics. For example, the decrease in the rate of surfaced larvae from region 3 between $IBM_0$ and $IBM_{LS}$ was related to smaller egg diameters. Nonetheless, deeper drifts brought valuable knowledge on several features that are developed below.

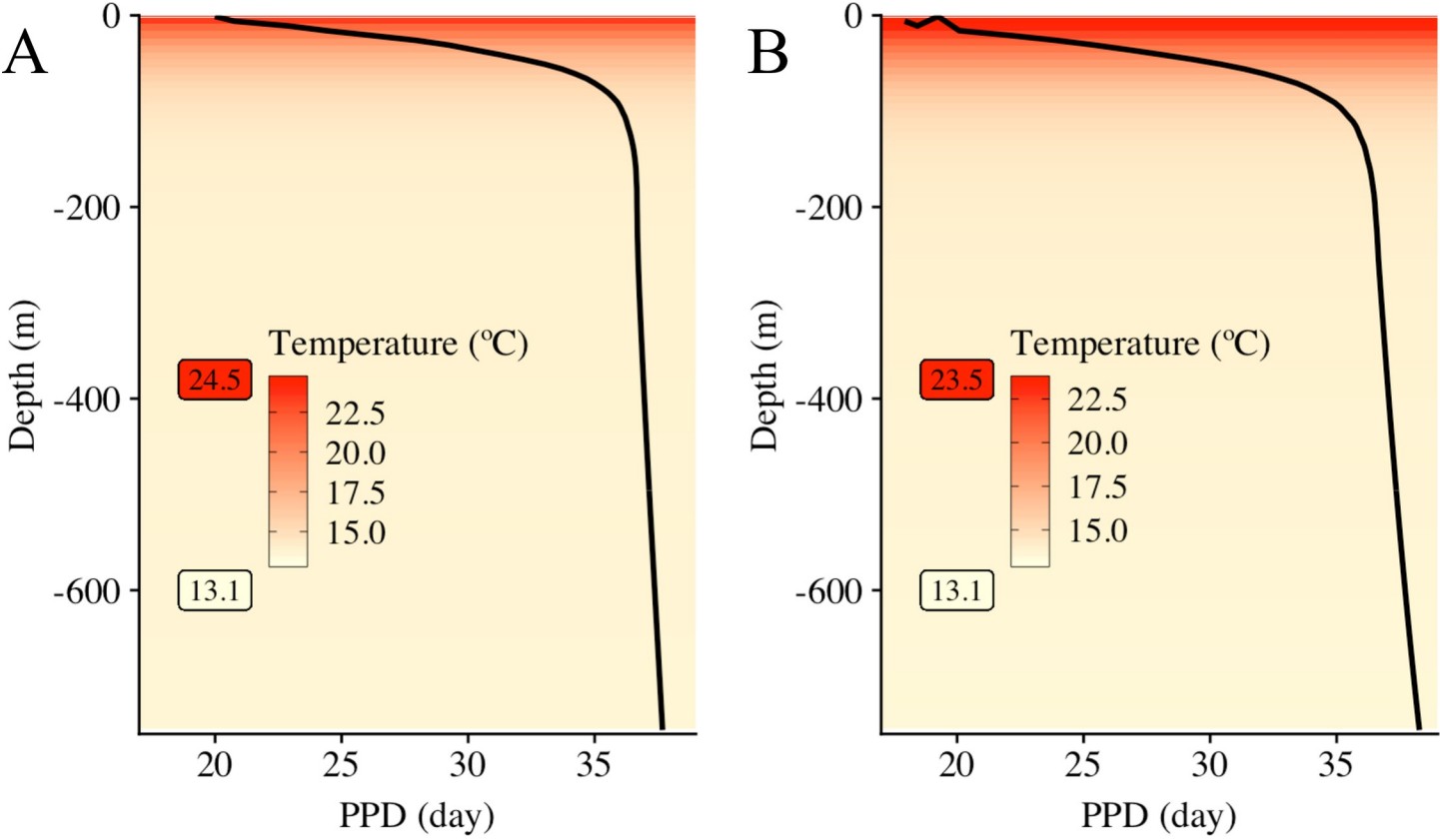

**Fig 8. Pelagic Propagule Duration (PPD) according to water temperature estimated at different depths.** The average duration of propagules (from eggs to early juveniles) of *Aristeus antennatus* is in relation with the decreasing temperature (color gradient,˚C) with increasing depths (Y-axis, m). In (A), PPD and water temperature from the IBM scenarios in early summer (IBM$_0$ and IBM$_{PZ}$) were averaged, and water temperature ranged between 13.1˚C and 24.5˚C. In (B), PPD and water temperature are from the IBM scenario in late summer (IBM$_{LS}$), where water temperature ranged between 13.1˚C and 23.5˚C. IBM scenarios are described in Table 1.

First, the water temperature gradient in different depth layers was advantageous for longer transports. Due to the variability in egg size and egg density, the buoyant stages stabilized at different vertical positions from zero to 75 m for more than 75% of individuals. The drifts were 20 km longer because they lasted 11 supplementary days in the deeper layers regardless of the scenario involved. The highest gradient in the drift duration occurred in the upper 100 m layer where the drift lasted 12 supplementary days at 100 m than at the surface (Fig 8). Beyond 100 m, the drift duration was stabilized at 36–38 days. Nonetheless, the scale of the deeper drifts by region showed that the circulation was still stronger and weaker in the region 1 and 2, respectively, where the drifts were the longest (167.5 km) and the shortest (102.5 km), respectively and regardless the scenario. In a concrete case, the simulated drifts from region 3 in IBM$_{LS}$ indicated that the upper and deeper layer were decoupled because individuals drifted two times longer than the individuals in the upper layer (Table 2).

Second, longer drifts and decoupled currents underneath the five meters depth favored the connectivity with the Balearic Islands (Fig 9). In other words, the velocity fields under the surface had slightly different directions. This was particularly seen by the drifts from the region 1 that, in two cases, provided higher arrival rates of individuals on the northwestern part of the Balearic Island grounds. One case is illustrated by the drifts in IBM$_0$ showing 33.1% to 40% individuals from the zones 2 to 4 in the 5–330 m layer depth were transported toward the islands. The second case is showed in the IBM$_{LS}$, with surfaced and deeper individuals

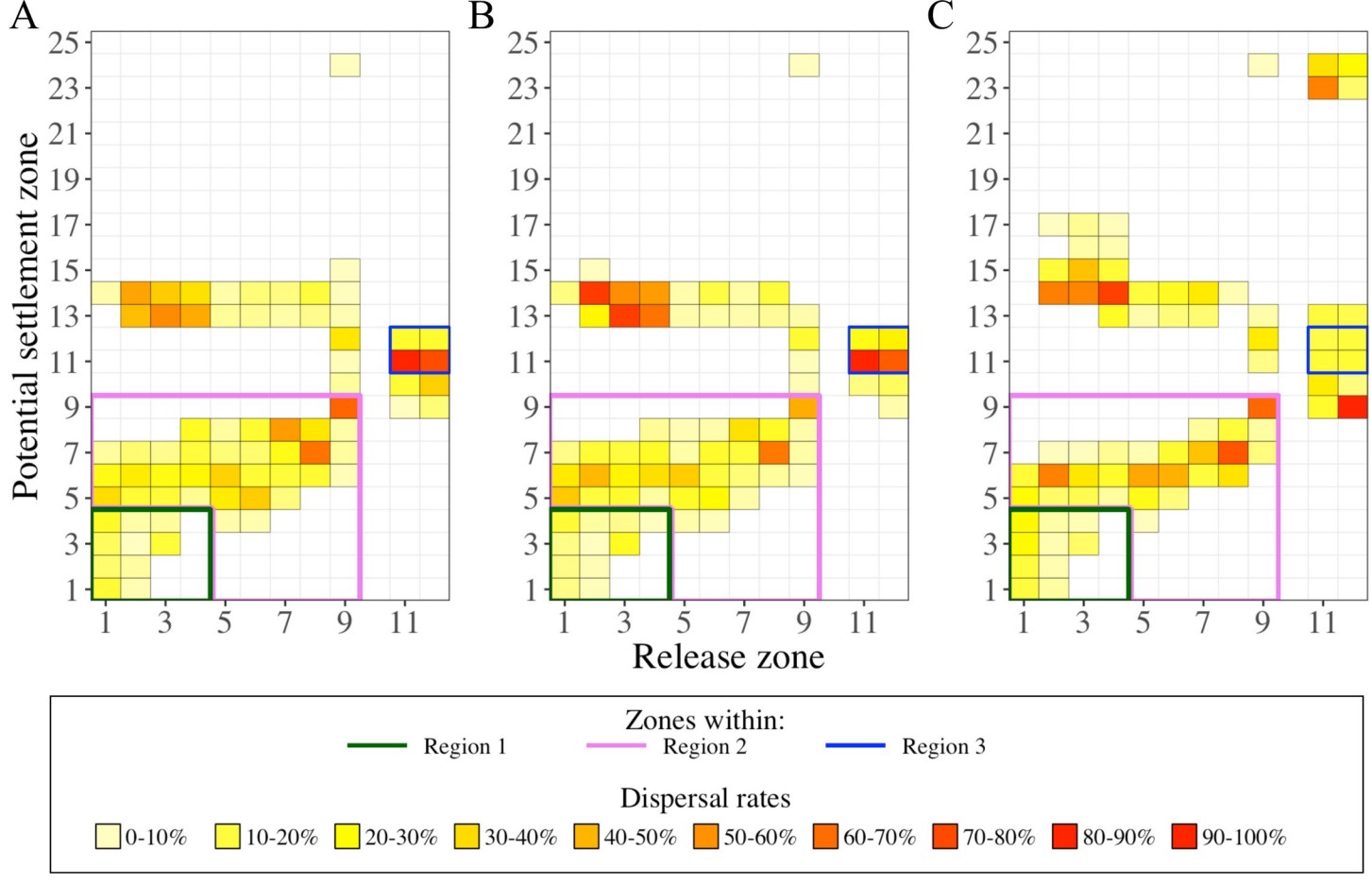

**Fig 9. Dispersal rate in the lower water layers (below 5 m) by the three selected IBM scenarios.** Dispersal rates were calculated from release zones (X-axis) to potential settlement zones (Y-axis) from simulated larval drifts in A) the base scenario $IBM_0$, B) the scenario with the buoyant stages up to Protozoea $IBM_{PZ}$, and C) the scenario initialized at late summer $IBM_{LS}$. For clarity purposes, we contoured the dispersal rates in regions 1 (green lines), 2 (pink lines) and 3 (blue lines). Identification of zones and regions as defined in Figs 1 and 4, respectively. Code color is related to the dispersal rate (%). The sum of the dispersal rate over a column gives the percentage of individuals below the surface layer.

reaching the zones 13 and 14, though the connection with zone 15 was established if only individuals were in the deeper layer (8 to 381 m depth). The same concept prevailed in the region of the Eivissa Channel. Within the first 100 m layer, longer drifts (160 to 400 km) from the Eivissa Channel connected to the south ground of the biggest Balearic Island (Majorca Island) by 39.5% individuals from zone 11 and 13.4% individuals from zone 12.

Finally yet importantly, the deeper and variable drifts of the individuals allowed broader connectivity. One release area could be connected up to eight other areas (Fig 9). Those connections were allowed by small numbers of individuals, which remained near the release areas, like for the case of individuals from region 1, which had a northwestward dispersal and thus, an opposite direction of dispersal than particles at the surface.

## Discussion

Model results emphasized the importance of the buoyant phases and their variability in the larval cycle of deep-sea species. In the upper water layer of the northwestern Mediterranean Sea,

the larval transport of *A. antennatus* relied on different circulation patterns and their spatio-temporal variability.

## Importance of buoyant phases

The simulations of different buoyant phases in the larval drift of the red shrimp *A. antennatus* emphasized the importance of the lecithotrophic stage in the drifts. In the recommendation guide for IBMs by [51], buoyancy is a key parameter for explaining the changes in the larval drifts. However, prior to our study in the Mediterranean Sea, only [15] studied the impact of egg buoyancy on simulated drifts for the deep-sea shrimp *Aristaeomorpha foliacea* (taxonomically close to *A. antennatus)*, using Atlantic anchovy egg density [52].

In our study, over half of the released individuals reached the layer where a high quantity of *A. antennatus'* larvae is found (i.e., the surface layer [11]). We estimated an average egg density of 884 kg/m$^3$, which was around 100 kg/m$^3$ lighter than the ones described for other deep-sea shrimp eggs (1025–1082 kg/m$^3$; [17]) and the seawater densities of the northwestern Mediterranean Sea (from 1025.3 kg/m$^3$ in Cap de Creus to 1024.1 kg/m$^3$ in the Eivissa channel; [23]). Nonetheless, egg density is partly species-dependent and therefore, egg density of deep-sea Decapod egg with bigger sizes than *A. antennatus'* oocyte [17] may not be appropriate to use in our Lagrangian model. However, the displacement of *A. antennatus'* eggs to the surface was comparable with some Penaeid eggs that were observed at the surface in laboratory water tanks and sea planktonic samples [50, 53], but this fact could not be quantitatively verified due to lack of Penaeid egg density data. To improve the next larval drift simulations with buoyant eggs of *A. antennatus*, the relationship between egg density and diameter (e.g., [17]) during the egg incubation should be considered. Indeed, when the eggs develop, several chemical reactions (i.e., the production of a perivitelline space by cortical activation of the eggs [54], the egg hydration [55], and embryonic development) likely modify the lipid rate, egg size, and egg density [53, 56]. Nevertheless, information about egg growth of deep-sea species and Penaeid eggs needs to be provided.

Extended to the nauplius stage, the rise of individuals illustrated the important role of the nauplius for increasing the probability to reach the surface layer. However, due to lack of knowledge, two parameters were not considered in the density values of IBM$_{PZ}$: the loss of weight at hatching and the change of shape at hatching [57]. In the Penaeid family, the elliptical and hairy morphology of the nauplius leads to a change in the buoyancy effect [58], which may speed up or slow down the ascent of the larvae after hatching. Furthermore, at this stage, previous studies provided the implication of the swimming behavior in the control of the larval vertical migration. Indeed, although erratically [20], [59] and [60] observed a rise of the Penaeid nauplii followed by a resting time in which the nauplii slowly sank. The velocity of *Penaeus* larvae is poorly informed although [20] suggested that larvae rise quickly after hatching, guided by phototaxis. Nevertheless, all authors emitted doubts about the ability of nauplii to swim up to the surface. We underline the necessity to carry out studies on the locomotion of decapod shrimp larvae in the open ocean to define the potential balance between the buoyancy of the nauplii and their swimming ability.

## Dispersal modulated by current variations

We noted that the main linear circulation along with eddy-like structures supported distinct dispersals of the red shrimp larvae according to their drift depths. Turbulence (in scenario IBM$_{Turb}$) or individuals at the thermocline depth (in scenario IBM$_{MLD}$) introduced small variations during the drift simulation that hardly modified the final dispersals of individuals. Dominant circulation patterns particularly controlled the larval dispersal rates according to

three regions in our study (instead of two in [61]). In region 1, the spatial distribution of larvae near the surface was strongly influenced by the Northern Current and winds, as reported in other drift analyses of marine larval species with egg buoyancy on our study area [62, 63]. In regions 2 and 3, the larval drift was limited by local mesoscale eddies, which have high importance in retaining larvae [64]. Near Blanes Canyon, the beginning of Ebro shelf, and the Eivissa Channel, previous studies identified eddies potentially trapping larvae of various species such as *Engraulis encrasicolus* and *Sardinella aurita* close to the North side of the Ebro shelf [65–69]. Decapod larvae have never been reported trapped in eddies in our study area, but [13] and [70] showed that decapod larvae were also aggregated by eddies south of the Balearic Islands and on the Norway lobster (*Nephrops norvegicus*) in the Irish Sea.

New connection or reinforcement of connection between zones by larval transport and resulted from the temporal changes of the eddy characteristics (i.e., if present, the diameter and the spatial position). During summer, the displacement and intensity changes in eddies along the Iberian Coast [65, 68] modified the simulated drifts, and therefore, the connectivity between areas. The drifts were influenced by a cyclonic eddy in the western part of the Eivissa Channel, as described previously by [71]. Its presence changed the larval dispersal by preventing the larvae from crossing the channel. However, during our late summer simulation, the fishing ground of *A. antennatus* near the Cabrera Harbor of Majorca Island [72], was supplied by individuals from the Eivissa Channel. The present study modeled and described for the first time how the Palamós canyon or the southern Eivissa Channel fishing grounds could connect to the fishing grounds of Majorca Island. Nonetheless, exploring the temporal evolution of eddies and their influence on the larval drift is needed to consider the possibility of persistent connectivity.

## Dispersals related to the location and position of larvae

Larval duration of surface *A. antennatus'* individuals depended to the Modified Atlantic Water mass dynamics. In our study, the PPD of larvae drifting at the surface varied along with a latitudinal gradient of near-surface temperatures of the NW Mediterranean Sea, which were colder in the vicinity of Cap de Creus and warmer at the Eivissa Channel [73]. This gradient has already been outlined by the interface of the old and cold Modified Atlantic water with the hot and recent Modified Atlantic water [74]. Additionally, it conveyed that larvae spawn from one of those regions will have distinct drifts from the other and partially explained the delimitation of the three zones in our study. The late summer dispersal enlightened a change of seawater temperature, which, besides the change in atmospheric conditions, can be related to the position of those two water masses. Indeed, the old Modified Atlantic water goes southward driven by wind and the southern entrance of the recent Modified Atlantic water is modulated by the water circulation in the channels of the Balearic Sea [74].

The larval drift in deeper layers can be a positive strategy induced by the buoyant stages. First, it implied that when the hatching occurred in deeper water layers, the PPD was longer. Second, the larvae drifted in the core of the North Current, which is maximal between 10 and 100 m [75, 76]. Third, the underneath surface is advantageous to larval dispersal because it may provide a good growth environment with nutrients located near the deep chlorophyll maximum [77] and protection from predators [28]. Fourth, the dispersal rate was wider and revolved round close settlement zones as well as far away zones. Consequently, for the first time, our study presents a potential connectivity between the two distant fishing grounds of Palamós and Soller (in Majorca Island from the Balearic Islands) areas and the expected drift conditions to link those places, where catches of *A. antennatus* are relatively important [72, 77].

## Conclusions

Our study analyzed the larval drift of *A. antennatus* under the hypothesis that either the nauplius stage or the protozoea stage is in the upper water masses. We considered concurrently the effects of two main factors on the connectivity that both contribute to the dispersal strategy of this benthic deep-sea species: buoyancy and water mass circulation. The buoyancy linked the eggs spawned at sea bottom to the larvae in the upper layers of the ocean. A consequence of the buoyancy was the split of the Lagrangian drifts into three latitudinal spawning areas in accordance with similar larval dispersal. Simultaneously, we found that connectivity patterns and retention rates were influenced by the presence of mesoscale structures such as meanders above the head of Blanes Canyon and eddies along the Valencian Gulf and in the Eivissa Channel. The drift of larvae in early and late summer followed the persistent main circulation pattern of the NW Mediterranean Sea, with temporally variable mesoscale structures adjusting the connectivity between the different zones. The application of buoyancy on different larval stages also highlighted that positions of the larvae in the water column influenced the dispersal direction and the intensity of connectivity between zones of the NW Mediterranean Sea. This study contributes to increasing the knowledge of *A. antennatus'* egg and larval ecology, and illustrates the potential dispersal paths of the individuals during their pelagic life. The spatio-temporal influence of mesoscale circulations in the larval dispersal should be tackled in further studies to better understand the red shrimp population connectivity, and to upgrade advice towards fishing practices and fisheries management.

## Supporting information

**S1 Fig. Decreasing Pelagic Propagule Duration (PPD) with increasing water temperature.** Continuous lines are the fitted exponential curves on reviewed data (cross) for each stage: eggs (orange), nauplius (red), protozoa (brown), and mysis (black). Reviewed data of PPD and temperatures were extracted from published articles and referenced in S2 File.
(TIF)

**S2 Fig. Number of particles needed to decrease the Fraction of Unexplained Variance (FUV) below the statistical threshold of 0.05.** The FUV expressed the bias created by a low number of particles (FUV > 0.05) after dispersal simulation. Whiskers show the minimum and maximum of the 100 FUV calculated for each number of particles tested (1000, 2000, 5000, 10000, 50000, 100000, 200000, 300000). Horizontal dashed line is the statistical threshold of 0.05.
(TIF)

**S3 Fig. Kernel density estimates from 50000 simulated individuals in different scenarios.** The simulated transport characteristics were A) the drift distances, B) the Pelagic Propagule Duration, and C) the water temperature during the drifts. Black lines indicate the scenarios grouped together by the PCA (Fig 3) with $IBM_0$, $IBM_{MLD}$, $IBM_{Diff}$ and $IBM_{Hot}$ in four different line types. Blue line indicates the scenario $IBM_{PZ}$, and grey line indicates the scenario $IBM_{LS}$. See Table 1 for a description of the IBM scenarios.
(TIF)

**S1 File. Information on eggs from deep-sea species and Penaeid species.** Deep-sea crustaceans (Decapoda) species and taxonomically close species to *Aristeus antennatus* with the depth ranges of adult distribution, egg diameter and egg density. Depth range extracted from Sealife Base (https://www.sealifebase.ca) or EOL (https://eol.org). * Studied species; *n/a*, non available data; in grey cells, species taxonomically close to *Aristeus antennatus*.
(DOCX)

**S2 File. Database on Penaeid embryonic and larval duration by stages.** T, water temperature (˚C).
(DOCX)

**S1 Table. Frequence of surfacing individuals (in percentage) in the preliminary experiments.** Percentage of buoyant stages reaching the surface or the Mixed Layer Depth (MLD) for a given density (kg/m$^3$) in Individual-Based Model modified by the release period (early summer and late summer) and by the activation of random turbulences.
(DOCX)

**S2 Table. Characteristics of eddies involved in the larval dispersals.** Three eddies were shaped by the larval dispersal simulation (see Fig 5). Coordinates of the eddy center were estimated with an algorithm from [34]. The edge of eddies was got from the streamlines. Computation of the eddy radius is based on the circle surface (Area = pi*Radius$^2$ and approximated the estimated eddy size.
(DOCX)

## Acknowledgments

The main author thanks all the Norwegian researchers from NIVA (Oslo, NO) who contributed to this paper: Johannes Röhrs, Knut-Frode Dagestad, Trine Bekkby and her family. The main author also thanks the editing from Hanna Morrissette and the advice from Elizabeth North from HPL-UMCES in Maryland (US). This study was possible thanks to the Spanish project CONECTA (CTM2014-54648-C2-1-R) lead from the ICM-CSIC in Barcelona (ES).

## Author Contributions

**Conceptualization:** Morane Clavel-Henry, Jordi Solé, Nixon Bahamon, Guiomar Rotllant, Joan B. Company.

**Data curation:** Morane Clavel-Henry.

**Formal analysis:** Morane Clavel-Henry.

**Funding acquisition:** Joan B. Company.

**Investigation:** Morane Clavel-Henry, Jordi Solé, Trond Kristiansen, Nixon Bahamon.

**Methodology:** Morane Clavel-Henry, Jordi Solé, Trond Kristiansen.

**Resources:** Jordi Solé.

**Supervision:** Jordi Solé, Nixon Bahamon, Guiomar Rotllant, Joan B. Company.

**Visualization:** Morane Clavel-Henry, Jordi Solé.

**Writing – original draft:** Morane Clavel-Henry.

**Writing – review & editing:** Morane Clavel-Henry, Jordi Solé, Trond Kristiansen, Nixon Bahamon, Guiomar Rotllant, Joan B. Company.

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
