## [Decision Letter · Decision Letter 0]

26 Nov 2019

PONE-D-19-26124

Modeled buoyancy of eggs and larvae of the deep-sea shrimp Aristeus antennatus (Crustacea: Decapoda) in the northwestern Mediterranean Sea

PLOS ONE

Dear Ms CLAVEL-HENRY,

Thank you for submitting your manuscript to PLOS ONE. After careful consideration, we feel that it has merit but does not fully meet PLOS ONE’s publication criteria as it currently stands. Therefore, we invite you to submit a revised version of the manuscript that addresses the points raised during the review process.

We would appreciate receiving your revised manuscript by Jan 10 2020 11:59PM. To enhance the reproducibility of your results, we recommend that if applicable you deposit your laboratory protocols in protocols.io, where a protocol can be assigned its own identifier (DOI) such that it can be cited independently in the future. For instructions see: http://journals.plos.org/plosone/s/submission-guidelines#loc-laboratory-protocols

A rebuttal letter that responds to each point raised by the academic editor and reviewers. This letter should be uploaded as separate file and labeled 'Response to Reviewers'.A marked-up copy of your manuscript that highlights changes made to the original version. This file should be uploaded as separate file and labeled 'Revised Manuscript with Track Changes'.An unmarked version of your revised paper without tracked changes. This file should be uploaded as separate file and labeled 'Manuscript'.

We look forward to receiving your revised manuscript.

Kind regards,

Atsushi Fujimura

Academic Editor

PLOS ONE

Journal Requirements:

Additional Editor Comments:

Please address all the comments, especially the Reviewer2's first point. Justify your choice of egg densities, or rerun the model with more realistic densities.

Reviewers' comments:

Reviewer's Responses to Questions

**Comments to the Author**

1. Is the manuscript technically sound, and do the data support the conclusions?

Reviewer #1: Yes

Reviewer #2: No

2. Has the statistical analysis been performed appropriately and rigorously? 

Reviewer #1: Yes

Reviewer #2: N/A

3. Have the authors made all data underlying the findings in their manuscript fully available?

Reviewer #1: Yes

Reviewer #2: No

4. Is the manuscript presented in an intelligible fashion and written in standard English?

Reviewer #1: Yes

Reviewer #2: No

5. Review Comments to the Author

Reviewer #1: The paper uses a circulation model for the eastern coast of Spain with estimated buoyancy and swimming for eggs and larvae of a deep spawning shrimp to calculate the potential recruitment areas for the adults.

The simulations start July 1 (with one starting September 1) making different choices for egg density and active processes. The major finding of interest is that there is a trade-off between temperature and drift distance (and direction) with slower transport at depth where cooler temperatures prolonged development in contrast to faster movement and development near the surface. The secondary effect is that the deep flow is often in a different direction from the surface flow leading to different recruitment location.

The primary determinant for the drift path was the egg buoyancy (L564) with an additional effect due to nauplius vertical motion, that may differ from that of eggs.

An issue with the analysis is that information is sparse on the character of eggs and nauplii of this species of shrimp during development. A good effort is made to use similar species, although there seems to be a bit of difference in characteristics among these species. In spite of this sparseness, the authors have made these choices clear so the reader can evaluate the results. My own interpretation of these results is that egg buoyancy is variable within a species and the model results inform the drift path of these shrimp based on this inherent variability.

A clear message from Fig 9 is that the Blanes/Palamós region provides abundant recruits to the Balearic Islands; this has management implications since the islands are not strictly self-recruiting.

Minor comments:

L175ff: It is important the the larval code includes both lateral and vertical turbulent effects, although these effects seem small compared to the persistent circulation in determining the final drift location.

L239: There is a difference in horizontal diffusivity being 100 here and 10 on L185.

L221: It would be nice to have identified the drift duration units to be days. These formula would apply for a constant temperature. But, the drifting organisms experience a range of temperatures. How is this information used? Also, these formulas might be more informative in the form D = Do exp(-a(T-To)) for some choice of base temperature (To). I am curious that all forms have the same e-folding temperature value (1/0.072 = 13.9 C). That value is close to the surface to bottom temperature difference (L152:153, 27.6 - 13.1 = 14.5). Given the large number of papers consulted, It seem odd that all forms would have the exact same coefficient. This might be that the Q10 values for all similar species is the same. Maybe the authors can comment on this result.

L395: I am not sure of the meaning of "The individuals from the region 2 shaped circular structures having different sizes." Are the eddies different sizes? Are the individuals different sizes? Are the individuals circular or spherical? Along the same lines, the area between the coasts and Islands have numerous eddies. How persistent are these flows? They likely are important in changing the flow path, so some statement of their size, strength and persistence would help the reader. Would the results be very different if the release time had been a week or two earlier or later than July 1?

Reviewer #2: To model the rising, transport and dispersion of the eggs and larvae, you must use background data as realistic as possible; data such as temperature, salinity, water-density, and current (motion of water) as physical properties, and egg/larva size, shape, and density as biological ones.

There are at least two fatal problems in this paper:

(1) The egg density smaller than 900 kg/m^3 is not realistic at all. Eggs are of protein, lipids and water (with ions). Lipids and water with ions in the eggs are less dense than ambient seawater, and thus provide the buoyancy. However, density of lipids is typically 900 kg/m^3, and proteins 1300. Therefore, 900 kg/m^3 for the egg density is entirely unrealistic.

I understand that you needed to adopt this unrealistically small density for the eggs to force the eggs spawn in the deep bottom to reach the surface. Does this mean that:

Because, without adopting such too small density, the model cannot explain the eggs to reach the surface from the bottom, there might be a possibility that the adult shrimp swim to the surface to spawn (which must be a so far unknown behavior)?

(2) Also, in Lines 152 and 153, I am afraid the values of seawater density you are showing might be wrong:

Average summer surface temperature, salinity, and seawater density from the ROMS outputs

are 27.6, 37.5 PSU, and 1026.3 kg/m3, respectively. Average summer bottom temperature,

salinity, and seawater density are 13.1 ºC, 38.2 PSU and 1035.35 kg/m3.

The seawater density as calculated from the temperature and salinity should be

T S density

27.6 37.5 1024.41,

13.1 38.2 1028.85.

Because you are using the wrong values for the egg density and seawater density, I am afraid your model, and the following discussions based on the computational experiments are invalid; therefore, this paper is not worth being published.

6. PLOS authors have the option to publish the peer review history of their article (what does this mean?). If published, this will include your full peer review and any attached files.

Reviewer #1: No

Reviewer #2: No

---

## [Author Response · Author response to Decision Letter 0]

9 Dec 2019

To whom it may concern,

A letter has been enclosed within the "attach files" section under the name of "Response to reviewers". You will have graphics and tables that could not have been included in this box.

The replies are associated with lines corresponding to the marked-up revised manuscript. If not, lines are indicated by n/a. We first responded to the Editor's comments, second, to the Reviewer 1 and last, to the Reviewer 2.

Regards,

Morane Clavel-Henry.

1) From Editor

Lines: n/a;

Comments: Please ensure that your manuscript meets PLOS ONE's style requirements, including those for file naming.

Answer: The manuscript has been modified to follow the template requested by the journal. These modifications do not appear in the marked-up manuscript.

Lines: n/a;

Comments: Please address all the comments, especially the Reviewer2's first point. Justify your choice of egg densities, or rerun the model with more realistic densities.

Answer: We justified our choice about egg density values as specified below in the reply to the reviewer. Manuscript was also modified at L. 279-281 and L.287-291.

2) From Reviewer 1

Lines: 175;

Comment: It is important the larval code includes both lateral and vertical turbulent effects, although these effects seem small compared to the persistent circulation in determining the final drift location.

Answer: We agree on this comment. In the preliminary scenarios, we considered the lateral and vertical turbulent effects (L.187-192). Simulation of larval drifts did not changed significantly from models without the turbulent effects as said around L.678-679. 

Lines: 271-273;

Comment: There is a difference in horizontal diffusivity being 100 here and 10 on Line 185

Answer: The maximum coefficient of horizontal diffusivity (i.e., 100 m2/s) was used in the experiment in order to identify the number of particles to be released. With this coefficient value, we reduced the risk to underestimate the number of particles. For clarifying the use of different horizontal diffusivity values (10 and 100 m2/s), we modified the manuscript L.270-274. 

Lines: 224-235;

Comment: It would be nice to have identified the drift duration units to be days. This formula would apply for a constant temperature. 

Answer: The duration units were in days. The unit was added L.246.

Comment: But, the drifting organisms experience a range of temperatures. How is this information used? Also, these formulas might be more informative in the form D = Do.exp(-a(T-To)) for some choice of base temperature (To). 

Answer: The organisms experienced a range of temperature that we considered by updating the stage duration according to the water temperature at the organism positions when the organism metamorphosed (L.253-257). For example, two protozoea had different stage duration if one of the nauplius molted in water at 14 ºC and the other in water at 23 ºC. This methodology was similarly used in the study of Phelps et al. (2015). 

Comment: I am curious that all forms have the same e-folding temperature value (1/0.072 = 13.9 ºC). That value is close to the surface to bottom temperature difference (L152:153, 27.6 - 13.1 = 14.5 ºC). Given the large number of papers consulted, It seem odd that all forms would have the exact same coefficient. This might be that the Q10 values for all similar species is the same. Maybe the authors can comment on this result.

Answer: In the manuscript, we adjusted a multiple linear regression to the dataset provided in Supplementary File 2. The regression model was performed to explain the stage duration in function of the water temperature and the stage regardless of the Penaeid species. In the earlier manuscript version, we split the result into four equations by stage for better vizualization. The multiple linear regression model explained a large amount of data variability (R2 = 95%) than single linear regression models fitted on individual stages (R2 between 31 and 59%) (Table A).

Table A. Linear regression models to estimate A. antennatus larval duration according to water temperature and two PLD estimations at 15ºC and 25ºC.

Because we were misunderstood, we added explanations about the multiple linear regression adjustment and grouped the Eqs [Disp-formula pone.0223396.e004] to 7 (see L. 250-258).

Lines: 445-452;

Comment: I am not sure of the meaning of "The individuals from the region 2 shaped circular structures having different sizes." Are the eddies different sizes? Are the individuals different sizes? Are the individuals circular or spherical? 

Answer: Eddies were with different sizes and shape, not the individuals. Modified in the manuscript at L. 437-445.

Comment: Along the same lines, the area between the coasts and Islands has numerous eddies. How persistent are these flows? They likely are important in changing the flow path, so some statement of their size, strength and persistence would help the reader. 

Answer: We provided further statements about eddy size and location with an eddy detection algorithm (L 435-451) described by Nencioli et al. (2010). This new reference is added in the manuscript, Lines 156 and the following S2 Table (L.441, 442, 444) was added as supplementary material. 

S2 Table. Characteristics of eddies involved in the larval dispersals. Three eddies were shaped by the larval dispersal simulation (see Fig. 5)

Comment: Would the results be very different if the release time had been a week or two earlier or later than July 1? 

Answer: We believe that different frequencies of individual release would have been different because to the residence time of the eddy (generally lasting for 2-3 weeks, Rubio et al., 2009). We consider that this topic is beyond the aim of the manuscript though we agree that studies in that direction should be done (L.706-707). 

3) From Reviewer 2

Lines: n/a;

Comment: To model the rising, transport and dispersion of the eggs and larvae, you must use background data as realistic as possible; data such as temperature, salinity, water-density, and current (motion of water) as physical properties, and egg/larva size, shape, and density as biological ones.

Answer: Modeling of eggs rising to the surface required knowledge about hydrodynamics, while biologic data would be necessary for validation of the density estimates and drift simulations. In our study, physical values (i.e., salinity, temperature, etc…) were provided by a climatologic hydrodynamic model implemented in ROMS and they were validated as indicated in Clavel-Henry et al. (2019). As indicated in the text (L.286), biological data (i.e., egg/larva size, shape and density) were inexistent. 

Lines: n/a;

Comment: Without adopting such too small density, the model cannot explain the eggs to reach the surface from the bottom, there might be a possibility that the adult shrimp swim to the surface to spawn (which must be a so far unknown behavior)?

Answer: This study simulated the rise of the first embryonic and larval stages of A. antennatus based on two statements, also provided in the manuscript (L.73-88). First, after several decades of studies/observation/fishery on adults of Aristeus antennatus, the deep-sea shrimp has never been seen/caught close to the surface. ROV diving caught the displacement of A. antennatus (see images: https://inpn.mnhn.fr/espece/cd_nom/538572) limited to few centimeters above the sea-bottom. Furthermore, several studies (Sardà et al., 1997; Tudela et al., 2003) assumed that the shrimp spawns from there. Second, collected larvae during planktonic surveys were caught in surface and identified at larval stages Protozoea I to Mysis II. The larval cycle is assumed similar to other Penaeid species: pelagic eggs, 6 nauplius stages, 3 protozoeas, 2-3 mysis. The earlier stages (eggs and nauplius) were not collected during the surveys. 

This is why we assumed that one or two buoyant stages (i.e., eggs or eggs and nauplius) crossed the water column, which separates the spawners depth and the first protozoea stage near the surface. Therefore, model estimations of egg density were required for simulating this displacement during the buoyant period.

Comment: The egg density smaller than 900 kg/m^3 is not realistic at all. Eggs are of protein, lipids and water (with ions). Lipids and water with ions in the eggs are less dense than the ambient seawater, and thus provide the buoyancy. However, density of lipids is typically 900 kg/m^3, and proteins 1300. Therefore, 900 kg/m^3 for the egg density is entirely unrealistic. 

Answer: As stated in the manuscript (L.73-88), we had an important lack of biological data about eggs and nauplius such as their size and density. To address this, we thought of three strategies indicated below:

1) To run the Lagrangian model, we intended to fill the gaps with information on eggs from taxonomically close species (i.e., Penaeids) and other deep-sea decapod crustaceans (see S1 File). However, the information on observed egg characteristics (e.g., size, size variation during incubation, and egg density) had too many gaps to be averaged (as for the estimation of the Pelagic Propagule Duration). For example, Penaeids egg densities are mentioned as comments about their vertical position in a rearing tank. Besides, the information also had too many differences to be directly used in our simulations. Another example is the case of Aristeus antennatus’ oocyte size, that is around 330 µm (Demestre and Fortuño, 1992), but deep-sea species and Penaeid egg size in S1 File measured between 145 to 4650 µm.

2) We intended to estimate the egg density from the wet mass of an A. antennatus oocyte and its diameter. In the thesis of Demestre (1990), average wet mass of oocytes could be estimated from the gonad weight and the number of oocytes measured from ready-to-spawn females (Table B). The egg density estimates ranged between 246 and 875 kg/m3. We hardly believed in the results because we could have underestimated the egg density by not omitting the weight of other tissues including in the gonad weight. In other words, the oocytes wet mass could not be correct.

Table B. Estimates of oocyte density. Measured ovaries were collected from the ready-to-spawn shrimp. Oocyte density was estimated from the oocytes wet mass and volume based on the average oocyte diameter at 330 µm.

3) We used the modeling method to simulate the displacement of individuals to the surface (see Fig. 4 in manuscript), which, in our case, was the best solution for approaching A. antennatus’ egg density. The model estimated that an individual egg density from 884 to 979 kg/m3 was needed for achieving the link between egg spawned at the bottom and first protozoea collected at the surface. We verified the results from numerical simulation with applied mathematics (i.e., using Sundby and Dallavalle equations) and calculated the egg density allowing the eggs to cross 500/800 m depth in 2 days (see Figure A). In the figure, at known Aristeus antennatus egg diameter (i.e., 330 µm), egg density were bracketed by [908, 975] kg/m3 (Figure A.A) and by [875, 962.5] kg/m3 (Figure A.B) to rise across 500 and 800 m depth, respectively.

Figure A. Egg density needed for the surfacing of eggs with different diameters. Egg densities were estimated in laminar flow (Re <0.5) and in transient flow (Re > 0.5) according to Sundby and Dallavalle equations, respectively. We used the average surface (1026.3 kg/m3) and bottom (1035.35 kg/m3) water densities and constant terminal velocities (0.0029 m/s, for eggs crossing 500 m depth in 2 days (panel A) and 0.0036 m/s, for eggs crossing 800 m depth in 2 days (panel B)). Range of eggs diameters used in the study is indicated by the grey polygon.

Given the value of the egg diameter and the hydrography of the Mediterranean Sea (i.e., warmer and saltier than other oceans), the egg density predicted to allow the surfacing of buoyant individuals approached the coherent values upper than 900 kg/m3. In our study, only the scenario with two buoyant stages (i.e., eggs and nauplii) estimated a density value upper than 900 kg/m3. Therefore, we can assume that the nauplius period is important for achieving larval dispersal in the upper layer (L.663-664). Otherwise, we can assume that other egg characteristics influence the vertical displacement. For example, during egg incubation, the egg diameter rises. Then, at constant water masses, a bigger egg can have higher densities (see Figure A). Nonetheless, these characteristics are poorly described for Penaeid species that have similar egg size with A. antennatus and we could not explore this possibility in the transport model.

To conclude on that point, we modified the manuscript L. 279-281 and L.287-291 to clarify our modeling decisions. We modified as well the discussion paragraph to underline the knowledge gaps in egg ecology that need further research at L.625-660.

Lines: n/a

Comment: Also, in Lines 152 and 153, I am afraid the values of seawater density you are showing might be wrong: Average summer surface temperature, salinity, and seawater density from the ROMS outputs are 27.6 ºC, 37.5 PSU, and 1026.3 kg/m3, respectively. Average summer bottom temperature, salinity, and seawater density are 13.1 ºC, 38.2 PSU and 1035.35 kg/m3. The seawater density as calculated from the temperature and salinity should be:

T S density

27.6 37.5 1024.41,

13.1 38.2 1028.85.

Answer: The seawater density value that we calculated is right as it corresponds to the average of the seawater density over the area of interest. The equation included as well the pressure like requested by 75-term equation for specific volume among the Gibbs function. We understood that the values suggested by Reviewer 2 were computed from the average values of temperature (e.g., 27.6 ºC) and Salinity (e.g., 37.5 psu) which is not representative of the average seawater density over the area. To avoid future misunderstandings, we modified the text at L.151-161 informing about how seawater density was averaged.

4) From authors

We added two more references in the manuscript at lines 154 and 156. The reference index was modified in the manuscript but this change has not been marked-up.

34. Nencioli F, Dong C, Dickey ., Washburn L, McWilliams JC. A Vector Geometry–Based Eddy Detection Algorithm and Its Application to a High-Resolution Numerical Model Product and High-Frequency Radar Surface Velocities in the Southern California Bight. Journal of Atmospheric and Oceanic Technology. 2010;27(3):564-579. doi:10.1175/2009jtecho725.1

35. Fofonof NP, Millard Jr RC. Algorithms for the computation of fundamental properties of seawater. UNESCO Technical Papers in Marine Sciences. 1983;44:53 pp.

References in the Rebuttal letter:

Demestre M. Biologia pesquera de la gamba Aristeus antennatus (Risso, 1816) en el mar catalán. PhD Thesis. Universitat de Barcelona. 1990.

Phelps JJC, Polton JA, Souza AJ, Robinson LA. Behaviour influences larval dispersal in shelf sea gyres: Nephrops norvegicus in the Irish Sea. Marine Ecology Progress Series. 2015;518:177-91.

Rubio A, Barnier B, Jordà G, Espino M, Marsaleix P. Origin and dynamics of mesoscale eddies in the Catalan Sea (NW Mediterranean): Insight from a numerical model study. Journal of Geophysical Research: Oceans. 2009;114(C6). doi: 10.1029/2007JC004245

Sardà F, Maynou F, Tallo L. Seasonal and spatial mobility patterns of rose shrimp Aristeus antennatus in the western Mediterranean: results of a long-term study. Mar. Ecol.-Prog. Ser. 1997: 133-141.

Tudela S, Sardà F, Maynou F, Demestre M. Influence of Submarine Canyons on the Distribution of the Deep- Water Shrimp, Aristeus antennatus (Risso, 1816) in the NW Mediterranean. Crustaceana, 2003;76(2): 217-225.

---

## [Editor Report · Decision Letter 1]

13 Dec 2019

Modeled buoyancy of eggs and larvae of the deep-sea shrimp Aristeus antennatus (Crustacea: Decapoda) in the northwestern Mediterranean Sea

PONE-D-19-26124R1

Dear Dr. CLAVEL-HENRY,

We are pleased to inform you that your manuscript has been judged scientifically suitable for publication and will be formally accepted for publication once it complies with all outstanding technical requirements.

With kind regards,

Atsushi Fujimura

Academic Editor

PLOS ONE

---

## [Editor Report · Acceptance letter]

14 Jan 2020

PONE-D-19-26124R1 

Modeled buoyancy of eggs and larvae of the deep-sea shrimp *Aristeus antennatus* (Crustacea: Decapoda) in the northwestern Mediterranean Sea 

Dear Dr. Clavel-Henry:

I am pleased to inform you that your manuscript has been deemed suitable for publication in PLOS ONE. Congratulations! Your manuscript is now with our production department. 

With kind regards,

on behalf of

Dr. Atsushi Fujimura 

Academic Editor

PLOS ONE